# Enhancing Microservices Security with Token-Based Access Control Method

**DOI:** 10.3390/s23063363

**Published:** 2023-03-22

**Authors:** Algimantas Venčkauskas, Donatas Kukta, Šarūnas Grigaliūnas, Rasa Brūzgienė

**Affiliations:** Department of Computer Sciences, Kaunas University of Technology, Studentu str. 50, 51368 Kaunas, Lithuania

**Keywords:** cybersecurity, microservices architecture, access control, external authentication, internal authorization

## Abstract

Microservices are compact, independent services that work together with other microservices to support a single application function. Organizations may quickly deliver high-quality applications using the effective design pattern of the application function. Microservices allow for the alteration of one service in an application without affecting the other services. Containers and serverless functions, two cloud-native technologies, are frequently used to create microservices applications. A distributed, multi-component program has a number of advantages, but it also introduces new security risks that are not present in more conventional monolithic applications. The objective is to propose a method for access control that ensures the enhanced security of microservices. The proposed method was experimentally tested and validated in comparison to the centralized and decentralized architectures of the microservices. The obtained results showed that the proposed method enhanced the security of decentralized microservices by distributing the access control responsibility across multiple microservices within the external authentication and internal authorization processes. This allows for easy management of permissions between microservices and can help prevent unauthorized access to sensitive data and resources, as well as reduce the risk of attacks on microservices.

## 1. Introduction

Microservices are an approach to software architecture that divides an application into small, independent, and self-reliant services [1,2]. The primary benefit of utilizing microservices is that it enables quicker development, easier maintenance, and enhanced scalability and resilience. In addition, microservices can increase the efficiency and flexibility of a development team by permitting the parallel development and deployment of individual services [3]. With microservices, it is also possible to utilize different technologies for various services, resulting in a system that is more flexible and adaptable. Microservices can result in faster and more efficient development, enhanced scalability and resilience, and greater organizational flexibility.

Nonetheless, one of the aspects that microservices architecture complicates is cybersecurity. Among distributed modules, it is difficult to ensure the confidentiality of data exchange and the integrity of transactions. The decentralized nature of the microservices architecture increases the number of potential entry points for attackers, making system security more difficult. A vulnerability in a single microservice can have cascading effects on the entire system; therefore, it is crucial to ensure the security of all services. Due to the distributed nature of microservices, real-time monitoring and detection of security threats can be challenging. Given the fact that misconfigured microservices can leave the system vulnerable to attack, it is crucial that all services are properly configured and secured. Microservices often rely on third-party libraries and services; therefore, it is crucial to ensure the security of these dependencies. Microservices architecture must prioritize cybersecurity and implement security measures, including encryption, authentication, access control, and monitoring, to mitigate these risks.

Communication between microservices can be complicated, making it more difficult to protect transmitted data. A set of smaller microservices must communicate with one another in a microservices architecture, so each microservice has its own communication interface. Obviously, the security of the microservices architecture can be implemented with a variety of perimeter security strategies, but other factors, such as action logging, performance monitoring, confidentiality of data exchange, and microservices availability, must also be considered. To enable secure communication in a microservices architecture, there must be distinct system boundaries. In a microservices architecture, there are two primary ways to define system boundaries. The first strategy is to assume that the application’s internal environment is secure, while the second strategy is to mistrust the environment and implement security mechanisms at the microservices communication level [4]. In general, the concept of secure communication encompasses not only the secrecy of requests or the execution of requests by an authenticated user, but also the assurance that the sender of the request is, in fact, authenticated to make this request. To be more specific, it is necessary to ensure that a microservice has permission to send a specific request to another microservice.

Access control plays a crucial role in ensuring the identity management and cybersecurity of microservices-based systems. Access control is an essential functional component of microservices architecture, as it regulates access to resources within each microservice and helps preserve the confidentiality and integrity of data. Access control in microservices helps to enforce security policies and prevent unauthorized access to APIs, data stores, and other resources. This ensures that only authorized entities have access to sensitive information. In addition, access control facilitates compliance with regulations, such as GDPR, HIPAA, and others, by controlling who has access to what data and for what purposes. This enables organizations to comply with regulations and avoid penalties for noncompliance. It is also an integral component of a comprehensive microservices security strategy and should be meticulously planned and implemented to protect sensitive data.

Centralized and decentralized access control are two different approaches to managing user access in the microservices architecture. Each approach has its own advantages and disadvantages, and the selection of which approach to use depends on the particular requirements of the microservices implementation. In centralized access control, all decisions regarding access control are made in a centralized location, making it simpler to manage and enforce security policies. In large-scale implementations of microservices, centralized access control can become a bottleneck and limit the system’s scalability. The reliance of centralized access control on a single service to control access creates a potential single point of failure in the system. Implementing centralized access control can be difficult, necessitating a high level of coordination between microservices and the access control service.

Decentralized access control allows for more scalable access control, as decisions can be made at the service level, and reduces the risk of a single point of failure, as access control decisions can be made by multiple services in the event of a failure. It can increase the complexity of the system, as each microservice must implement its own access control logic [5]. The decision between centralized and decentralized access management depends on the specific requirements of the microservices’ implementation, such as the system’s size and complexity, the required level of security, and the need for scalability and resilience. There is no single “best” method for token-driven decentralized or centralized access control in microservices, as the specific requirements and constraints of implementation for each microservice dictate the strategy that needs to be adopted.

For the client and the user, authentication and authorization serve slightly different purposes [6]. The purpose of the application system in the case of a user is to verify his or her identity and grant him or her access to the relevant resources and functions. In the case of a client, however, the concept of authorization is typically more expansive due to request quotas. Requests from external customers from other organizations are frequently monetized and subject to time-bound limits. In a microservices architecture, it is more difficult to implement restrictions on the number of requests because a single client request can generate multiple new requests to other microservices; therefore, it is essential to define the request quota policy accurately.

A modern practice to track and limit client requests is to use special access tokens or session identifiers. It is important to note that the client may not be a person but another computer system that uses the functions of another organization’s application system. JWT tokens can be used as an access control mechanism in the microservices architecture [7]. Tokens are a standardized method of representing authentication and authorization data, and they can be used to implement a wide variety of access management scenarios in the microservices architecture. Tokens enable decoupled authentication, where authentication and authorization data are stored in a separate service and can be easily shared between microservices. Tokens are stateless, which means they do not store any data on the client or server. This reduces the risk of security incidents and simplifies the management of session data. However, when validating a JWT token, it is essential to ensure that the cryptographic signature component exists, as a JWT token without a cryptographic signature is also valid. In this way, the attacker has the ability to include any information in the message, thereby gaining greater unauthorized access to the system.

Unfortunately, JWT-based access control is vulnerable to security issues, including interception, forgery, replay, expiration, storage, and size. To mitigate these issues, token-based access control systems should implement token signing and encryption, use short-lived tokens, and securely store tokens on the backend, *etc*. In addition, access control systems should routinely validate tokens and revoke them as needed to maintain security. Due to this, the objective of this work was to create a method for access control that ensures the enhanced security of microservices. The authors focused on the process of access control by evaluating and comparing it over centralized and decentralized architectures of the microservices. The research issue highlights the ongoing need for a secure functional architecture of microservices that can effectively prevent unauthorized access. The main contributions of the work, presented in this paper, are as follows:The microservice access control method is based on a token strategy that increases security in access control by distinguishing between authentication and authorization services, the issued tokens of which are trusted by the resource microservices. This ensures that the authorizations of the users between microservices are easily managed.External authentication and internal authorization provide enhanced security in client’s identity management while accessing the microservices.

The field of microservices has been growing rapidly in recent years, and with it comes the need for enhanced security measures. Although existing research has suggested a variety of techniques for securing microservices, many of these techniques have drawbacks, like slow performance or restricted flexibility. Compared to existing research, the proposed method offers several advantages. First, it provides a high level of security by generating unique tokens for each microservice request; thereby, preventing unauthorized access. Second, it is flexible and can be customized to meet the specific security requirements of different microservices applications. The efficiency of our approach can also speed up microservices requests by obviating the need for repeated authorization and authentication checks.

Future researchers who use this paper can benefit from the latest techniques in microservices security by implementing our proposed method. By doing so, they can enhance the security and performance of their own microservices applications, while also building upon our research to further advance the field. Additionally, the proposed method can serve as a starting point for exploring new approaches to microservices security and improving upon existing methods.

The remainder of this paper is organized as follows. Section 2 reviews the methods and solutions used for access control in microservices. The detailed description of the proposed token-driven access control method is provided in Section 3. Section 4 outlines the environmental setup for the experimental testing, while the results and insights based on the testing are presented in Section 5. A summary of the main findings and how they relate to the research issue is discussed in Section 6. Finally, Section 7 concludes the work done by the authors in this paper.

## 2. Related Works on Access Control in Microservices

Appropriate access control in the architecture of microservices is critical for assuring system security and integrity, as well as preventing unauthorized access to sensitive data and resources [8]. Access control helps to prevent data breaches or other security incidents and minimizes the risk of data corruption or tampering. It ensures that all microservices are properly configured and secure, as well as authenticated and authorized, which can help prevent attacks based on compromised credentials or minimize the attack surface of the system. On the same point, it can enforce consistent security policies and compliance with industry and government regulations across all microservices, which can make it easier to manage and maintain the security of the system. Besides all this, it reinforces the isolation of microservices from each other, preventing a security breach in one microservice from spreading to others [9].

Microservices are a paradigm that combines ideas from software engineering’s principles of simplification, usage, and division of concerns, as well as service orientation and distributed system concepts. This combination results in brand new security issues, in addition to the same old security challenges, but presented in brand new packaging [10]. The microservices paradigm, which consists of highly distinct and easily redeployable distributed components, also presents new potential for data security. The authors in [11] focused on security implications related to the deployment of microservices in a cloud environment. They concluded that, in order to provide users with secure microservices and accomplish a variety of other productive goals, it is necessary to pay attention to critical aspects of developing architectures that are based on microservices and the control of access to them. Such findings were justified as well by other researchers in [12].

Typically, in a microservices’ architecture, essential functions can be divided into several different levels of security, depending on the type of the architecture. In a layered microservices architecture, where each microservice is designed as a modular and self-contained component, with a specific responsibility, all microservices on the same subnet have the same level of security. The necessity of cross-layer connectivity and various bridgeheads, in terms of a security, was highlighted for a multi-layered microservices architecture in [13]. The authors identified that a need to improve security requirements for IoT microservices in the supply chain and data sharing systems still remains. Microservices have different levels of security and may require additional validation or authorization before certain critical functions can be performed.

Access control in microservices refers to the mechanisms used to regulate and manage the access and permissions at edge level, service level and context of identity of different services and users of the system’s resources [14]. In the model of an access control of distributed architecture of microservices, proposed by the authors in [15], the private API gateway is only available to the internal microservice, the so-called front-end of microservices. It takes care of the targeted invocation of critical microservices. This front-end calls the internal API gateway, which then sends the request to the filter service. The filter service is an additional tool to ensure the validity and authorization of the request. The filter object is separate from the authorization server, as the filter should only implement the additional business logic that needs to be implemented before critical microservices can be reached. The filter can also perform additional authentication by contacting the authorization server. If successful, the filter sends a request to the microservices with the next security level. This model follows the principle of a trusted security perimeter, which, in principle, requires additional measures to ensure the security of the sub-network. Such a type of architecture is more flexible and easier to scale than monolithic [16,17] architecture, as each service has its own separate databases and communicates with other microservices over the network.

It is important to note that security issues that arise in a monolithic architecture can be mitigated by deploying and scaling microservices independently, implementing proper access control, and proper orienting (i.e., centralized, decentralized, distributed or hybrid) authentication and authorization of systems and services. Segmentation helps as well in enforcing access control, by isolating microservices, so that unauthorized access to one microservice does not compromise the security of others [18]. The solutions of microservices segmentation were published in [19,20]. In terms of security, the objective of microservice segmentation is to limit the attack surface by isolating services. In this manner, if a single service is compromised, the damage is isolated to that service and does not propagate to the rest of the system. Additionally, microservices may be divided according to the degree of access they need, with more critical services requiring more stringent security measures. This contributes to the confidentiality, availability, and integrity of the data handled by the services.

A centralized authorization technique for microservices architecture is the role-based access control (RBAC) concept. It sets roles and gives permissions to those roles in order to regulate access to microservices’ resources. In this manner, each user is allowed access depending on their role, and authorization choices are made using this data. Nonetheless, the authors in [21] indicated that RBAC has several security concerns, including role expansion and division of duties. These concerns also impact the microservices’ security practices. In this instance, they suggested attribute-based access control (ABAC) for a centralized microservices architecture. ABAC provides a more dynamic and adaptable approach to access control than standard Role-Based Access Control (RBAC) systems. In an RBAC system, access is allowed based on a user’s assigned responsibilities, while ABAC considers the user’s characteristics and the request’s context. However, implementing ABAC in microservices introduces obstacles, such as attribute management complexity, performance cost, difficulty establishing consistent rules, lack of standardization, and security threats related to a wrong policy setup. The extended Role-Based Access Control model (Hierarchical Trust RBAC) for decentralized microservices was proposed in [22]. To guarantee that none of the cross-domain queries might lead to Cross-Site Request Forgery (CSRF) or Cross-Site Scripting (XSS) attack, the chain of trust in each service is tied to the user. In order to guarantee the reliability of microservices, this proposal does not provide two-factor authentication or advanced encryption technologies.

For access control, it is important to implement secure communication, which includes not only the confidentiality of requests or the execution of requests by an authenticated user, but also ensures that the sender of a request is indeed authenticated to make that request; in other words, it ensures that a microservice has the right to send a particular request to another microservice. In a microservices architecture, access control between the user and the microservices can be realized with an internal JSON Web Token (JWT) model [23]. Such a model ensures that each request is authorized by the internal service that issues the tokens. In a token-based strategy, the microservices architecture generates a token after the user has been authenticated and granted permission. The token is then transmitted with each request to the microservices, enabling them to determine the user’s identity and authorization level. Contrary to normal practice, such signed tokens are not returned to the user; in other words, the token’s life cycle is equal to the request life cycle. This security measure ensures that an unauthorised request coming from the outside is not executed.

However, the implementation of an internal JWT token strategy can cause several problems [4]. First of all, because of the communication rate, if a user sends a large number of requests to an API gateway in a short period of time, the overall response time of the system would decrease dramatically. The question then arises: Is it really necessary to issue a new token every time? A potential solution to this problem could be token caching at the API gateway or at the user authentication service. In the case of caching, the expiration date of tokens should be taken into account. Tokens that have expired or are about to expire can be removed from caching. Secondly, if a shared secret [24] (in the case of symmetric cryptography) or a public key [25] (in the case of asymmetric cryptography) is used to validate a token, this needs to be distributed between systems. It is inefficient to create a separate certificate issuer that all microservices can trust. Therefore, it would be possible to apply the attribution during an automated deployment. However, a change of key in the token service would require a re-installation of all microservices that validate the token, due to the change of key value.

In decentralized microservices, this is not an easy task, as it would probably mean that the system would not be fully accessible to the user at the time of a shared key change while microservices access management solutions were implemented. To allow an application system to access the resources of another system, the OAuth2 authorization protocol can be used [26]. The study, published in [27], suggested that OAuth2 is a widely used protocol and concluded that it helps applications manage user identities in a standardized way. The OAuth2 protocol is sufficiently broad and flexible that the authorization scenario can be implemented in several different ways. However, it is important to note that the complexity of the system tends to increase the possibility of errors and, thus, the likelihood of vulnerabilities, so it is always recommended to use the simplest possible model. There is no exception to this for the implementation of an identity management model using the OAuth2 protocol. Another proposed OAuth2 authorization model, in [28], presents interconnection between OAuth2 protocol and RBAC. Each scope describes the context that is available to the user and the functions and constraints that are applied to the user. This provides flexibility at the application level, where the same role has different access to the user depending on the context. The basic authorization scenario checks that the login credentials match the data in the system. If the data match, a token is generated, which includes the roles belonging to the user. The generated token is returned to the client. The client can then make a request to the application to access a specific resource by adding the previously issued access token. It is important to note the validity period of the token and the revocation of the token. It is recommended to issue tokens with the shortest possible expiration time to increase security.

It is important to note that interoperability with other systems is one of the purposes of OAuth2-based access control and the evaluation of this criterion may imply a sophisticated approach to identity management between different services. However, in the case of internal tokens, the complexity of the security mechanism is not straightforward, as the topological network schema has additional identity management services. It should not be difficult to solve the compatibility problem either, as internal tokens are not directly provided to the client, thus “hiding” the complexity of identity management.

In terms of perimeter security, in the OAuth2 model, the first request to a resource is always authenticated, but, depending on what is done next with the request, whether further authorization is implemented depends on the use case. In the case of internal tokens, perimeter security is unavoidable, because service communication is precisely based on trust in these tokens. Monitoring activities is one of the most significant features of a secure system. In a layered microservices architecture, [29], this is relatively easy to achieve as microservices are divided into layers where each layer serves a distinct role and provides a different level of abstraction. In this way, all requests are authorized in one place.

On the other hand, the OAuth2 model can also be quite simple if you have a single authorization service. The more authorization services there are, the more complex the action log becomes. The internal token model complicates tracking because it is not clear where to, and how, the request goes after authorization. Therefore, each service must ensure the correct auditing of actions. The layered approach has the fastest speedup because it validates a request only once during its lifetime. In the case of OAuth2, an authorization service is commonly used, which must be invoked before accessing the resource, so the client must take care of the authorization itself. For internal token instances, the speedup is the worst, because separate authentication and authorization services are used. In addition, each request receives an additional internal token that is trusted by all internal services. Issuing this token, and sending it for each subsequent request, incurs additional time costs and reduces the overall network bandwidth, depending on the token and the amount of information transmitted.

The main access control strategies considered in the microservices architecture are different in their own ways. It is not possible to answer the question, “Which is the best access control strategy?”, as each has its own pros and cons and use cases. The question should therefore be: Which access control strategy is most appropriate in a particular case? To answer this question, it is necessary to compare the strategies against each other (Table 1).

However, the strategy of controlling access with internal tokens is probably the most secure approach. Unlike the layered or OAuth2 models, internal tokens do not “issue” an authorization structure to clients. Internal tokens also provide perimeter security, making the system “borderless”. In other words, such an application system is protected against unauthorised access attacks from an internal subnet to unprotected resources. However, the medium token strategy is inferior to OAuth2 when it comes to integration with other systems, as the medium token strategy does not have an agreed-upon authorization protocol, in contrast to the OAuth2 model. Internal tokens have to carefully address the issue of bandwidth, as issuing and transmitting internal tokens with each request requires additional time and network bandwidth costs.

## 3. Method for Token-Driven Access Control

The authors’ proposed method is based on an internal JWT token strategy, in which each microservice validates a request using an additional token issued by the authorization service specifically for the incoming request (Figure 1). The implementation of an additional external token to be delivered to the client serves as a supplement to the token strategy used inside. This token is utilized in the client authentication process. With a valid authentication token, a client is able to send a request to the application system. However, before the request can be executed, it must be authorized by appending an additional access token containing information about the services to which the client has access. When the request containing the internal token arrives at the service, the service must validate the token. In addition to validating the token’s validity, the validation includes authorization for the required services.

It should be noted that only registered users have access to the system of microservices. When logging in, a user who has chosen to enable two-factor authentication (2FA) during account management is required to enter a login code provided by the access control subsystem. This is an additional way of securing against unauthorized access to the user’s account, which is now considered standard practice. The external access token is given to the user who is in charge of it. This token must contain the basic information required to establish the user’s identity and, optionally, basic information that does not fully describe the user’s privileges in the system.

The user sends a request to access a microservice, which is first processed by the access control subsystem. If the microservice is open to the public and does not require authorization, the request is simply routed to it. Otherwise, when authorization is required, the access control subsystem validates the user’s external access token received at login. After that, an internal access token is generated and attached to the user’s request, which is then forwarded on. Other microservices use this internal token to authorize the user’s actions.

In a microservices architecture, communication interfaces play a critical role in enabling independent service development and deployment. In this approach, the communication between microservices is enabled through RESTful APIs. REST APIs use HTTP and are language-agnostic. This means of the communication is employed for the following security-focused reasons: (1) REST APIs are stateless, which means that they do not store any session information on the server, reducing the attack surface area; (2) it allows for granular access control to specific resources, allowing for authentication and authorization at the resource level; (3) REST APIs can be secured through the use of encryption protocols, such as SSL/TLS, which can help protect data in transit; and, finally, (4) it is based on standard HTTP methods and status codes, making it easier to implement security measures and ensure interoperability. The external communication channel is determined by the implementation, but it could be an email system, a text messaging service, or something else. This external channel sends a validation code to the authentication service, which the user enters. This is an additional authentication step that aids in the prevention of unauthorized access. Depending on the request, a response is returned to the user once it is verified that the authentication code generated and the one provided by the user match. This means that the response to the user differs depending on the use case. Depending on the specific implementation, error messages are displayed to the user if an unforeseen event occurs at any step.

Such a strategy for access control in microservices has the following advantages:structure of external and internal strategy of JWT tokens;processes of external authentication and internal authorization;secure boundaries of the system;simplified recording of actions during the access;management of access control rate.

The use of internal tokens inevitably creates a distinction between the tokens for the system’s internal applications, which are used in inter-service requests, and the external ones, which are used by the client to send a request to the system. This token management allows the internal token structure to be separated from the external token structure, and, thus, hiding from the client certain internal authorization details, such as user identification data, request waiting times, fine access details, or other variables that are only relevant to the application system at the time of processing the request.

The use of internal tokens ensures the existence of authorization security between the microservices. In other words, each request must be validated so that the problem of “public access” does not create additional security risks. This is particularly convenient when services are to be deployed in different environments on servers between which it is difficult to ensure the security of the outer perimeter. If a malicious person breaks through the outer perimeter, his or her access is still limited, due to the strict policy of validation and authorization of requests.

External authentication and internal authorization ensure that the client is authenticated once and that each request is authorized by one service. This means that microservices that receive an internal request only have to validate the request itself. This reduces time costs, as otherwise each service would have to authenticate and authorize the user separately.

Although there may be a large number of microservice applications in the system, user authentication and authorization are only carried out at the relevant services. This means that it is not only convenient to record the fact that a client has been identified but also the access rights granted to him. Additionally, a unique request identifier can be added, which exists inside the internal token. This identifier could allow linking the action log of the authorization service with the action logs of the microservices, thus facilitating tracing not only when users have been authenticated or authorized, but also which specific actions have been performed by their queries.

Probably the most important shortcoming in microservice access control is speed. The additional time cost of access control for each client request can lead to relatively high latency times, especially if the client sends a large number of requests in a short period of time. However, this problem can be adequately handled in the proposed approach. In particular, it is possible not to send the whole request to the authentication service, but only the essential part of the request related to the identity of the customer. The next step is to split the access control service into authentication and authorization services. This evenly distributes the load required for access control. As far as the authorization policy is concerned, it is possible to grant a user the right to make the same request if the same request has been successfully authorized before the appropriate time period has elapsed. It is also possible to increase the validity of the request depending on the use case. Both the average total processing time of a request in the application system and the typical volume of user requests in a given time period should be considered. Thus, it is possible to increase the speedup in these ways, but with caution, as both of these solutions have a negative impact on the overall security of the application system.

The authorization microservice’s purpose is to generate an internal token that is used by other system microservices later in the request’s lifetime. If the authentication microservice’s validation of the external token issued to the user fails, the authorization microservice must respond to the query manager with an error message. Otherwise, if the external token is successfully validated, a message is sent, along with the generated internal token, which contains detailed information about all of the user’s rights. It is worth noting that it is possible to insert not only information about the user privileges obtained from the database, but also aggregated system information that would be present in the subsequent authorization of the request, during the creation of the internal token. For example, if a user requests that the email microservice send emails to all users, the authorization microservice may add a restriction to the list of privileges limiting the user to only performing a certain number of actions within a given time period. This restriction may vary, depending on factors such as system occupancy, time of day, and other external circumstances. Since the same constraint can be used in multiple resource microservices, a constraint that controls the speed of actions must belong to the authorization microservice. As a result, such a feature is extremely beneficial when the system performs functions that are influenced by external factors. Adding a token to these constraints allows for a more flexible solution to this problem.

The query manager is an important intermediary between the user and the system in centralized access control in microservices (Figure 2 and Figure 3).

The query manager attaches the internal token generated during the authorization process to the query, which is then forwarded to the appropriate microservice of the resource. The external token issued during authentication is a simplified information aggregate that is handed over to the user’s control. The user must send this token, with subsequent requests, in order for the authorization microservice to be able to successfully identify on whose behalf the request was sent. The authorization microservice creates an access token that contains the detailed user information necessary for the successful execution of the request on the resource. Finally, the resource microservice authorizes the user and, based on the internal token, executes the user request.

Such an access control approach requires three main microservices: a query manager, authentication, and authorization. The client applies only one of them, the query manager. If the client wants to authenticate, they contact the query manager, which forwards the request to the authentication microservice. It issues the necessary external tokens and returns them to the query manager, and the query manager returns them to the client. The query manager forwards requests to other requested resources as well, but if the requests require authorization, an additional request should be sent to the authorization microservice for an internal access token. The request to be authorized must be authenticated, i.e., have a valid external access token in the header. On receipt of the response, the request manager forwards the request with the additional authorization token further to the business subsystem. The resource microservices autonomously validates the internal access tokens and autonomously decides whether to authorize a client action.

In the decentralized access control method there is no query manager and each microservice authorizes each request independently (Figure 4). This means that if a single client request generates multiple internal requests in the internal microservice subsystem, then each microservice individually contacts the authorizing microservice and generates an additional stream of authorizing requests. In other words, a client request is always authorized only once in centralized authorization, whereas it is authorized several times in decentralized authorization, depending on the number of internal requests generated.

The proposed access control approach uses the JWT token in the microservices architecture, which has the key advantage of statelessness. Therefore, an internal or external token issued during the access control process remains active until it expires. It is, therefore, very important to define the validity periods of the different tokens. The next important aspect is the specification of the external token. Despite the fact that the external token is essentially a single entity, it can be split into several independent parts. In the case of JWT, the external token could consist of access and renewal tokens. The external renewal token would be sent to the authentication microservice with the purpose of obtaining a new access token when the old one expires. Meanwhile, an external access token would be sent to the system for authorization purposes.

Three JWT tokens are used in the proposed access control method: external access (Figure 5a) and external update (Figure 5b) tokens, and an internal access token (Figure 6). External tokens are given to the user and, therefore, have a higher chance of being damaged. External tokens should, therefore, use longer keys and signatures. However, the internal token should be used between microservices with a short expiry time at the system’s perimeter, so a shorter key and signature can be used, taking into account the high-speed performance. Therefore, the EdDSA elliptic curve algorithm can be the signature algorithm for the JWT token, which has high speed performance [30]. The Ed448 (448 bits) and Ed25519 (255 bits) versions of the elliptic curve signature algorithm can be used for external tokens and internal tokens, respectively. The validity of the tokens depends on the specific implementation, but the internal access token has the shortest validity (for the lifetime of a single request) and the external update token the longest.

The JWT tokens used in the proposed access control include *kid*, *sroles*, and *claims* fields, in addition to the standard *iss*, *sub*, *aud*, *exp*, *iat*, and *jti* fields [31]. The key identifier is needed to determine which symmetric key to validate the token with. As tokens can be issued with different keys at different points in time for the same JWT, this field ensures that the tokens are validated with the key with which the token was issued. However, this requires additional tracking of the key identifier. Nevertheless, this strategy allows the system to rotate keys (e.g., to replace a potentially compromised key) without significant inconvenience to customers: at the same time, customers are issued tokens with the new symmetric key, while tokens already issued are validated with the old one, until all tokens expire. It also increases the security of the system by allowing a set of keys to be used and rotated on a regular basis, which makes it more difficult for malicious parties to use cryptographic analysis techniques to predict the secret key used by the server by analyzing the token.

Access control is more flexible if a user can have multiple roles. These roles are not required for the external update or internal access token, but are required for the external access token. Based on the roles of the entity, the resource microservice knows the basic capabilities of the entity and allows or denies the user to perform certain actions in the system based on these capabilities. The authorization field is mandatory for the internal access token as it specifies detailed authorization information related to the entity’s capabilities in each microservice. This information may be partially provided in the external token if the use case requires it. A field for request conditions is also distinguished, which details the external system conditions for the request that are not directly dependent on the user. This could be, for example, speed limits for a particular microservice, due to an already high system load, or similar.

## 4. Environmental Setup for Experiments

The technology package selected to implement the access control method in the microservices architecture is based on Microsoft tools. The *Azure Service Fabric* platform was chosen to implement the method’s prototype because it provides the infrastructure required for managing, deploying, communicating, and extending the system for microservices. The prototype of the access control method in the microservices architecture has seven microservices that work in a common set and form an access control mechanism that can be used by other software developers (see Table 2 and Table 3).

The authentication microservice has the dual function of connecting the user to the system by issuing external access and update tokens. The microservice only provides these tokens after the user has provided the correct login and password. The prototype implementation assumes that the user’s login is his or her email address and is unique. The authentication microservice can also update the external access token upon presentation of an update token by the user. Thus, when the authentication microservice receives a valid updated token, it issues new access and update tokens that allow the user to continue to access the microservices of the resource and successfully use the application system.

Microservices communicate with each other, generating internal traffic that mirrors the way the real system works. Each resource function is protected by an additional authorization that verifies the internal access token sent in the HTTP request header *MicroAC-JWT*. The actual microservices of the resources, their URLs, their functions, and the necessary permissions, are defined by the system developers, who integrate the proposed access control approach into their application system. The developers need to additionally configure the request manager to accept requests for the various resource microservices. The microservices themselves also need to be implemented in such a way that they accept an internal access token and validate it with the same key that was used in the issue procedure. Once these integration aspects have been realized, software developers are able to make full use of the proposed microservices-based access management approach.

The query manager, authentication, authorization, and resources’ microservices provide additional request marking. This marking can be enabled or disabled by specifying the appropriate parameter in the configuration of the microservice. The marking is performed at the beginning and at the end of the request processing. This means that each microservice marks each request at least twice, immediately on receipt and just before sending the response. The tagging process is the addition of metadata to the *MicroAC–Timestamp* header of the HTTP request. The data to be added is the date, time, name of the microservice, the status of the request, and the action performed. This information can be used to analyze what specific actions were performed during the processing of the request.

The hardware, which was used during the experimental testing of the proposed method, had the following technical parameters:Windows 10 64-bit operating system;Intel Core i5-6600K processor with 4 cores and 3.50 GHz clock speed (6th generation, production in 2015);24 GB of RAM, type is DDR-4, and speed 1330 MHz;Azure Service Fabric Runtime 8.1.329 and SDK 5.1.329;Microsoft SQL Server 2019 (15.0.2080.9).

The experimental research aimed to evaluate the proposed access control method’s speedup on various criteria—CPU load, RAM usage, amount of external and internal query processing, query execution time—and to compare its implementation in centralized and decentralized architectures of the microservices. The experimental research focused on sending the requests from the client workstation to the workstation where the *Service Fabric* cluster was deployed, together with a complete prototype of the access control method within the microservices architecture. Both workstations were connected to the same local network. The access control prototype also used a database that was installed on the same workstation. The workstation also ran a hardware monitoring tool that collected information on CPU and RAM usage and wrote this information to a file. Integration tests were also run before each speed test to ensure that the cluster had been installed successfully and that the application system was ready for the speed tests. The cluster had 5 nodes, and one microservice replica was installed on each node. This meant that the workstation was running 35 separate prototype microservice processes (7 microservices on 5 nodes) simultaneously.

Speedup testing verified the performance of the prototype using centralized and decentralized modes of operation, together with optimized and non-optimized communication. For optimized communication, a mechanism was developed that uses *Service Fabric* libraries to compile, cache, and use the addresses of all microservices in the microservices’ cluster during all tests. This avoided the need to contact the *Reverse Proxy* server of *Service Fabric* for each request; thus, reducing the overall request traffic in the cluster and saving resources of the workstation. However, each microservice used a specially designed software class integrated into the microservice. This was suitable for parallel execution of requests and eliminated the need to use an additional element in the cluster, which overloads the network.

The experimental tests included a total of 12 speedup tests consisting of 4 modes with 3 different loads. One speedup test was programmed to execute a specified number of threads that continuously sent requests to existing microservice API functions. The tests used a constant number of threads, which depended on the load:the low load carried 1 thread;the average load carried 3 threads;the high load carried 6 threads.

The number of threads for each load was selected according to the results of the intermediate experiments, which showed that 6 threads were the maximum load that could be handled by the method’s prototype implemented on the workstation. A thread can be interpreted as a client that sends requests sequentially without interruption and waits for a response. This strategy generates a linear load on the application system, so this test simulated a steady stream of requests from customers. The duration of the chosen test was 20 min. In the intermediate experiments, it was observed that this was a sufficient time period to detect anomalies and ensure consistent results. Thus, the total minimum duration of the experimental testing to collect data was 312 min, or 5 h and 12 min (12 tests of 26 min each).

After each query was executed, the header tags generated by each microservice that contributed to the processing of the query were collected. The essential information in the tags is the name of the microservice, the node number to which the microservice belongs, and the start and end times of the processing of the request in the microservice. After 20 min, when all requests had been processed, a tag analysis algorithm was executed, which was individually programmed to investigate the speedup of the method’s prototype.

## 5. Results

CPU usage indicates (Figure 7) that the centralized mode used fewer CPU resources than the decentralized mode. At low load, the centralized mode was almost twice as efficient as the decentralized mode. For medium and high loads, the average difference between centralized and decentralized modes was 5% of the processor load. It could also be observed that, for both medium and high loads, the non-optimized mode used, on average, 6% less CPU resources.

However, given that the non-optimized communication consumed more RAM (Figure 8) and processed fewer requests at a slower rate, it could be assumed that the lower CPU usage did not mean a more efficient communication model, but rather that the reverse proxy waited longer for queries to be answered by the internal microservices, and, thus, made inefficient use of the CPU resources.

Looking into the processing of the queries (Figure 9), the centralized authorization mode used the query manager microservice, while the decentralized mode did not. The centralized mode compensated for this difference with fewer requests to the authorization microservice. However, the increased number of authorization requests in the decentralized mode meant not only a higher consumption of workstation resources, but it also enhanced security in request management. Unlike the query manager, the authorization microservice additionally accessed the database and searched for data related to user roles and permissions. During high load, the authorization microservice processed almost 100,000 more requests in decentralized mode than in centralized mode, which represented almost 40% of all internal requests in the mode.

In the optimized decentralized mode, the authorization microservice received, on average, 60% more requests than the centrally managed one.

The first two criteria, CPU and memory usage, describe hardware usage. It is also important to consider the use of physical resources to assess the effectiveness of a method. Figure 10 shows the weighting factors for each of the criteria.

The criteria and their qualitative indicators were normalized into values and multiplied by a weighting factor. The first normalization formula for each criterion can, therefore, be expressed as $x_{norm}$:(1)$$x_{\mathrm{norm}}=\left(1-\frac{x_{i}-min\left(x\right)}{max(x)-min(x)}\right)\cdot k$$

The normalized values with weighting factors were then summed to give an overall estimate of the test variation with the formula:(2)$$s=\sum_{x_{norm}}^{5}$$

The final normalization formula for each criterion can, therefore, be expressed as $v_{norm}$:(3)$$v_{\mathrm{norm}}=10\;\frac{s_{i}-min\left(s\right)}{max(s)-min(s)},$$

After applying the formulae (Equations (Equation 1)–(Equation 3)) to the experimental results obtained, it was found that the proposed access control method was, on average, the most efficient with low (score of 8.27 in the scale from 0 to 10) and medium (score of 6.94) loads, but its higher security was realized in a decentralized microservices architecture (score of 7.77). Decentralized access control can enhance security in access control of microservices by distributing the access control responsibility across multiple microservices. This reduces the risk of a single point of failure or attack, as well as minimizes the impact of a breach or compromise of one microservice on the rest of the system. External authentication in decentralized microservices refers to the use of external JWT tokens in order to authenticate users before they can access the microservices. This approach ensures that only authorized users can access the system and helps to prevent unauthorized access. Internal authorization in decentralized microservices refers to the use of separate microservices or components to manage access control for different parts of the system. This approach allows for fine-grained access control, where permissions are granted based on user roles and responsibilities [32]. By separating authorization from the main microservices, the risk of unauthorized access to sensitive information or functionality is reduced. In comparison to centralized access control, where all access control responsibilities are handled by a single component, decentralized access control provides a more distributed and secure approach to access control. A centralized approach is more vulnerable to attacks and may create a single point of failure for the entire system. With decentralized access control, each microservice handles its own access control responsibilities, reducing the impact of a breach or failure in one component on the entire system. Generally, decentralized access control with external authentication and internal authorization provides a more secure and scalable approach to access control in microservices.

## 6. Discussion

The microservices architecture access method typically uses stateless tokens to provide centralized access control to authentication and authorization services. These tokens are issued by the authentication and authorization services and are trusted by resource microservices. This allows for easy management of permissions between microservices and ensures that users have the necessary permissions to perform specific actions. The access control strategy is based on user right templates that define sets of rights with permissions to perform specific actions. By using a model of minimum privileges, the system is more secure, and access management can be more flexible for users.

One advantage of microservices architecture is its ability to scale horizontally by adding more microservices to the system. This means that the microservices can be developed and deployed independently, allowing for greater flexibility and scalability in the system. The microservices architecture solves several problems, but one of the main problems it addresses is the monolithic architecture’s limitations in terms of scalability, flexibility, and agility. In a decentralized microservices architecture, the processing of internal authorization is typically more secure than in a central authorization system. This is because each microservice is responsible for its own security and authentication, rather than relying on a central authority. This can help to prevent unauthorized access to sensitive data, as each service is able to verify the validity of incoming requests before processing them. The use of JWT tokens for access control in microservices architectures can be related to both technical and organizational aspects of security.

From a technical perspective, the use of JWT tokens can help to provide a secure and efficient way of managing authentication and authorization across different microservices. By using JWT tokens, it is possible to ensure that only authorized clients are able to access the microservices, and that each microservice is able to verify the validity of incoming requests. This approach to security can help to prevent unauthorized access to sensitive data and resources, and can also help to reduce the risk of attacks, such as cross-site scripting (XSS) and SQL injection. From an organizational perspective, the use of JWT tokens can help to promote a culture of security within an organization. By implementing robust security measures, such as JWT tokens, an organization can demonstrate its commitment to protecting its assets and sensitive data. This can help to build trust with customers and stakeholders, and can also help to attract and retain top talent in the industry.

Every research study has limitations as well. The paper focused on a limited set of microservices applications, and the proposed token-based access control method does not generalize well to other types of applications or contexts. Therefore, the findings of this study necessitate further research to evaluate the method’s effectiveness in different application domains. While the researchers conducted 12 speed-up tests to evaluate the proposed method’s performance, these tests did not consider the impact of different network topologies, or the effects of latency, on the method’s performance. Therefore, additional testing and evaluation may be necessary to fully assess the effectiveness of the proposed method. Token-based access control is only one aspect of microservices security, and there are additional security risks that the suggested method might not be able to address, like SQL injection attacks or cross-site scripting attacks. Therefore, while the proposed method may enhance security in some respects, it may not provide a comprehensive solution to all security threats.

In conclusion, the proposed token-based access control method could improve the security of microservices, but there are a few things to keep in mind when judging how well it works and how well it fits needs. Future research could address some of these limitations and further refine the proposed method.

## 7. Conclusions

With a monolithic architecture, all the application’s functionality is built into a single database, making it challenging to scale and maintain as the application grows. Microservices architecture, on the other hand, breaks down the application into small, independent services, each responsible for a specific task or function. This makes it easier to scale and maintain the system, as individual microservices can be updated and deployed independently of each other. This article suggests a way to control access that gives the client a simple authentication token and gives internal microservices an authorization token. By making changes to external and internal access tokens, you can hide details about authorization that may be different for each client. The use of JWT tokens can also promote collaboration and communication between different teams within an organization. By using a common approach to access control across different microservices, teams can work together more effectively and efficiently. This can help to reduce the risk of errors and vulnerabilities in the system, and can also help to improve the overall quality and security of the system. By implementing robust security measures, such as JWT tokens, organizations can improve their security posture, build trust with customers and stakeholders, and promote collaboration and communication between different teams. Moreover, the use of stateless tokens for centralized access control to authentication and authorization services makes it easier to manage permissions between microservices and ensures that users have the necessary permissions to perform specific actions, making the system more secure. In conclusion, the use of JWT tokens for access control in microservices architectures can be related to both technical and organizational aspects of security.

After looking at, and adding up, all of the criteria, such as the use of hardware resources, the number of requests processed, and the lengths of those requests, the proposed centralized access control mode is twice as efficient as the decentralized one. When only the optimized modes are taken into account, however, the difference in how well the centralized mode works grows seven times. The average difference in processor load between centralized and decentralized modes for medium and high loads was 5%. It was also observed that the non-optimized mode utilized, on average, 6% fewer CPU resources under medium and high loads.

Going further, the 12 tests that were performed showed how well the proposed token-based access control method worked. Regarding the latest applications influencing security issues, more recent and diverse applications were considered in the research in order to provide a more comprehensive evaluation of the proposed approach. Additionally, including challenges specific to these latest applications added more context to the research and provided insights into the current state of microservices security. However, expanding the number of tests, or including more recent and diverse applications, could be a direction for future research.

## Figures and Tables

**Figure 1 sensors-23-03363-f001:**
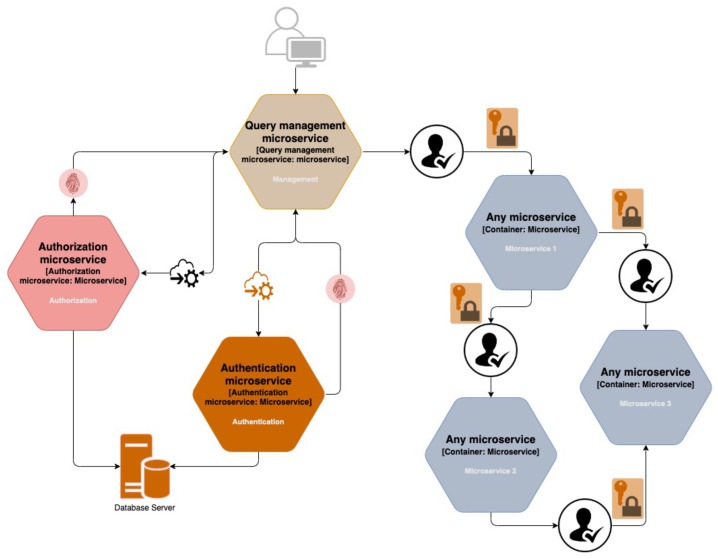
Concept of the JWT-driven access control in microservices.

**Figure 2 sensors-23-03363-f002:**
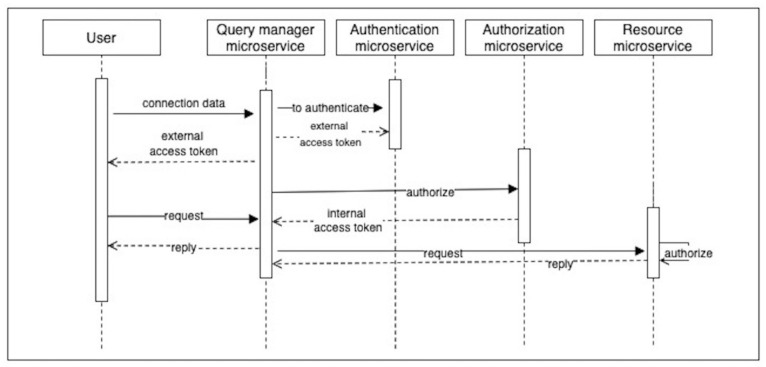
UML sequence diagram of the generalized process for centralized access control in microservices.

**Figure 3 sensors-23-03363-f003:**
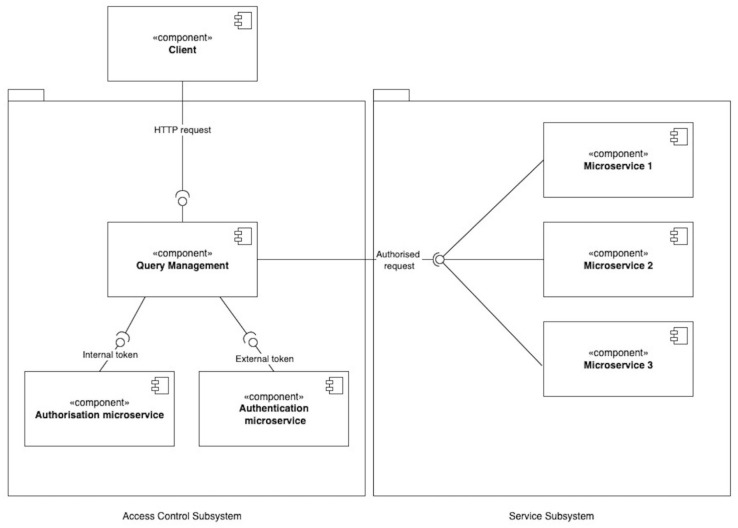
Diagram of the components for centralized access control in microservices.

**Figure 4 sensors-23-03363-f004:**
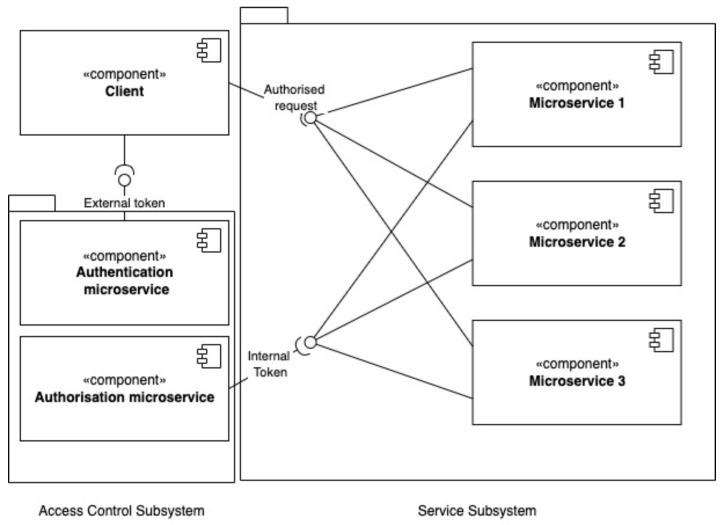
Diagram of the components for the proposed decentralized access control in microservices.

**Figure 5 sensors-23-03363-f005:**
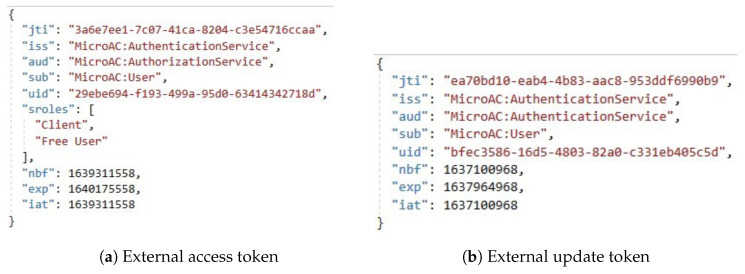
Structure of external JWT tokens.

**Figure 6 sensors-23-03363-f006:**
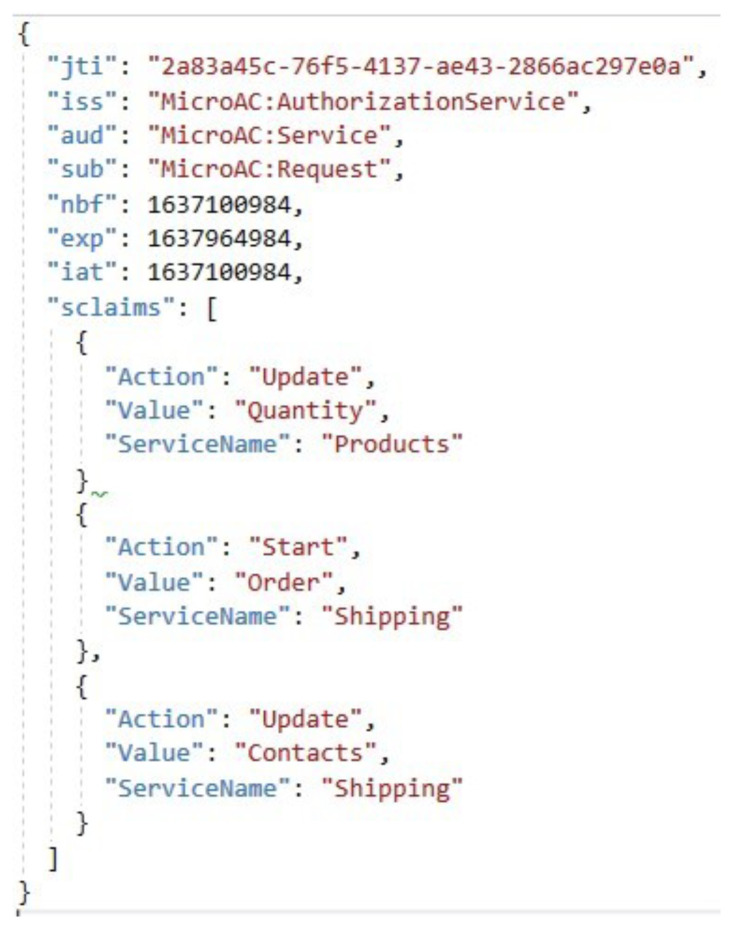
Structure of internal JWT token.

**Figure 7 sensors-23-03363-f007:**
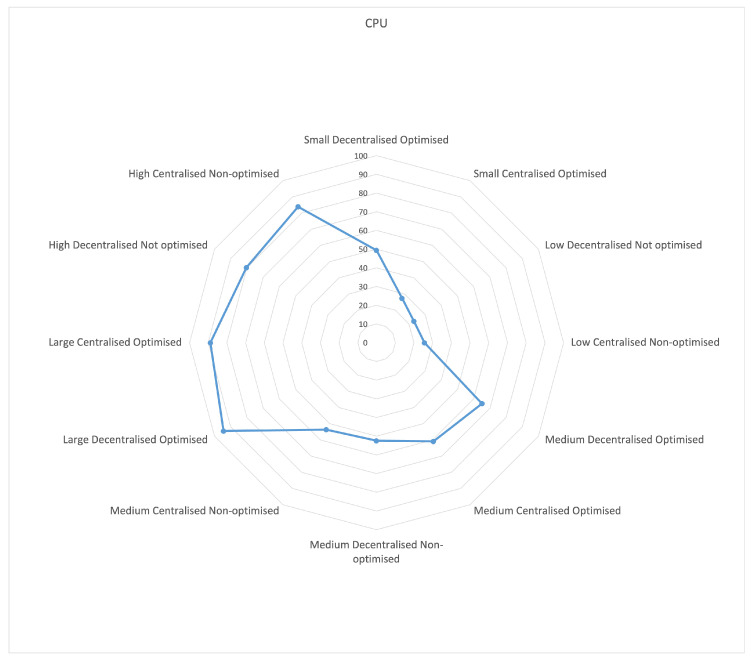
Usage of CPU.

**Figure 8 sensors-23-03363-f008:**
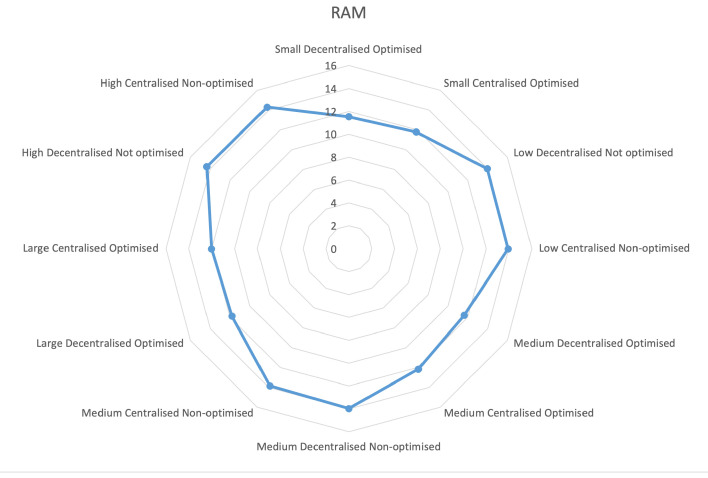
Usage of RAM.

**Figure 9 sensors-23-03363-f009:**
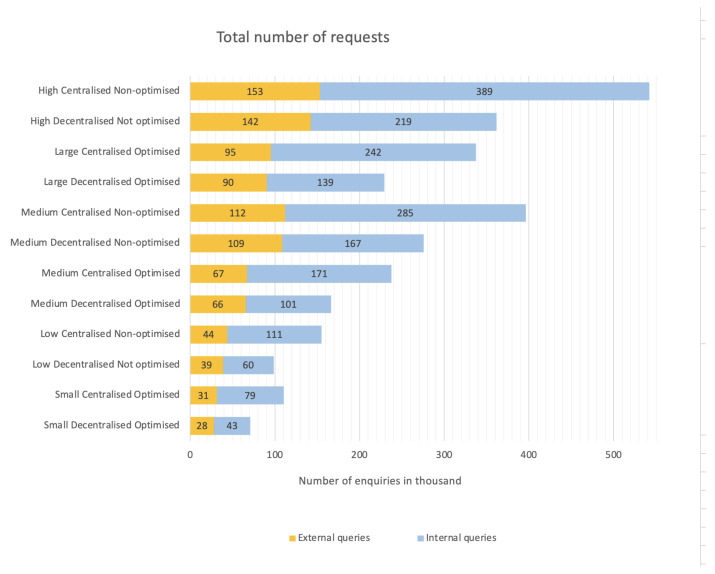
Processing of requests.

**Figure 10 sensors-23-03363-f010:**
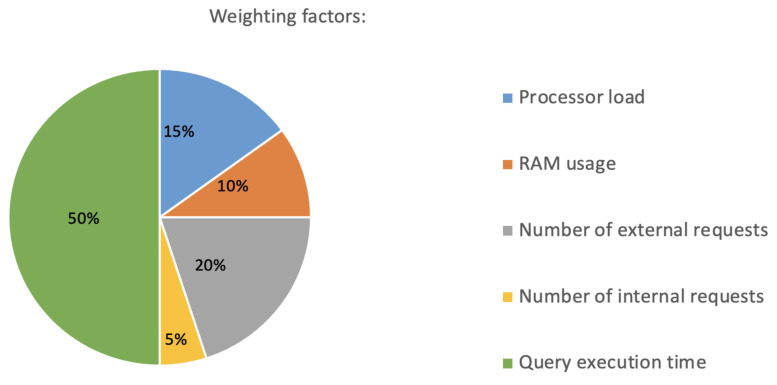
Weighting factors.

**Table 1 sensors-23-03363-t001:** Comparison of the access control strategies.

Method	Advantages	Limitations
Token-based	Unique tokens, flexibility	Limited evaluation, no comparison with other methods
RBAC	Granular control	High complexity, difficult to implement
ABAC	Contextual control	High complexity, difficult to implement
OAuth	Widely used, supports delegation	Complex, may lead to security vulnerabilities
OpenID Connect	Supports identity federation	Complex, may lead to security vulnerabilities
JWT	Portable, flexible	No built-in revocation mechanism

**Table 2 sensors-23-03363-t002:** Specification of query manager, authentication, authorization of the microservices’ API function.

Microservice	No.	URL	HTTP Method	Input	Function
Query manager	1	*/RequestManager*	Any	Client request	Receiving, auditing, and forwarding the internal token of a request
Authentication	2	*/Authentication/Login*	POST	Client login and password	Issuing an external access token
	3	*/Authentication/Refresh*	POST	Update token	Updating an external access token
Authorization	4	*/Authorization*	POST	External access token	Issuing an internal access token

**Table 3 sensors-23-03363-t003:** Specification of microservices’ API function.

Microservice	No.	URL	HTTP Method	Input	Function
4 business microservices	5	*/*	GET	None	Get the microservice
	6	*/{id}*	GET	Microservice ID	Get the microservice
	7	*/*	POST	New microservice	Create a microservice
	8	*/{id}*	PUT	Existing info of microservice	Update the microservice, apply to microservice_*n*_

## Data Availability

The data presented in this paper are available on request from the corresponding author. The data are not publicly available due to the project not being completed.

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
