# Peer review of "Enhancing Microservices Security with Token-Based Access Control Method"

_sensors, 2023, doi:10.3390/s23063363_

Round 1
Reviewer 1 Report
The mean idea of the paper is to propose a control access method, for microservices, baed on JWT tokens. The authors suggest to give the client a simple authentication token and give internal microservices an authorization token.
The Introduction and the related works are well presented. However, to improve the quality of the paper, I suggest that the authors draw up a comparative table at the end of the "related works" section.
Authors should also justify the choice of Azure Service fabric and how the management of certificate and public keys is done?
Author Response
Many thanks for the reviewer’s positive comments and the careful suggestions.
Please see the attachment.

Reviewer 2 Report
The paper is overall interesting and fits within the scope of this journal (Enhancing Microservices Security with Token-Based Access Control Method). This paper presents the enhancement of microservices security.
1. The authors proposed an interesting approach for enhancing microservice security using Token-Based Access Control Method.
2. The authors used 12 speed-up tests to prove the enhancement of microservice security.
Using more latest research work, authors may add a comparison table at the end of section 2 and compare with at least 5 to 10 techniques with appropriate parameters.
The introduction needs some modifications such as the difference between the proposed and existing research and quick benefits for future researchers who use this paper with the latest techniques.
In lines 41 to 43, the interface is used in that sentence. Although it is an interesting point to this research, the interface used in this research is not clear or explained anywhere in this paper and can be elaborated with the security issues. I think, from the security point of view, it may be discussed with the microservices.
It seems that there is no discussion about the limitation of this work. The authors could provide a detailed discussion of the limitations, to provide a thorough understanding of the property of the proposed method.
In Section 4, line 485 says that the microservices architecture has 7 microservices but in Tables 1, and 2, there are 8 microservices. How? I think the authors have to explain Tables 1 and 2 properly.
The authors have chosen and used 12 speed-up tests. Why 12? According to the latest information, many microservice applications are considered. Authors may add some challenges regarding the latest applications influencing the security issues and the maximum number of tests.
The conclusions can be improved by adding some numerical comparisons mentioned in the results. Future work may be added to the conclusions and results obtained from this research.
References are ok but the 22 references may not be enough so, authors may add more references.
Author Response

(The authors gave the same response as above.)
